# How Long Do Implanted Triclosan Sutures Inhibit *Staphylococcus aureus* in Surgical Conditions? A Pharmacological Model

**DOI:** 10.3390/pharmaceutics14030539

**Published:** 2022-02-28

**Authors:** Frederic Christopher Daoud, Ruben Goncalves, Nicholas Moore

**Affiliations:** 1INSERM U1219, Bordeaux Population Health, Bordeaux University, 146 rue Léo Saignat, CEDEX, F-33076 Bordeaux, France; ruben.goncalves@u-bordeaux.fr (R.G.); nicholas.moore@u-bordeaux.fr (N.M.); 2CHU de Bordeaux, Laboratoire de Pharmacologie et Toxicologie, Place Amélie Raba Léon, CEDEX, F-33076 Bordeaux, France

**Keywords:** sutures, triclosan, 3Rs, in vitro model, release kinetics

## Abstract

(1) Background: Sutures with triclosan (TS) are used to reduce the risk of surgical site infections (SSI), but most clinical trials are inconclusive. The traceability of SSI risk to antimicrobial activity in operated tissues is needed. (2) Objectives: This study aimed to predict triclosan antistaphylococcal activity in operated tissues. (3) Methods: Three TS were exposed to static water for 30 days, and triclosan release was recorded. Polyglactin TS explanted from sheep seven days after cardiac surgery according to 3Rs provided ex vivo acceleration benchmarks. TS immersion up to 7 days in ethanol-water cosolvency and stirring simulated tissue implantation. Controls were 30-day immersion in static water. The release rate over time was measured and fitted to a predictive function. Antistaphylococcal activity and duration were measured by time-kill analysis with pre-immersed polyglactin TS. (4) Fifteen to 60-fold accelerated in vitro conditions reproduced the benchmarks. The rate prediction with double-exponential decay was validated. The antistaphylococcal activity was bactericidal, with TS pre-immersed for less than 12 h before then *S. aureus* began to grow. The residual triclosan level was more than 95% and played no detectable role. (5) Conclusions: Polyglactin, poliglecaprone, and polydioxanone TS share similar triclosan release functions with parametric differences. Polyglactin TS is antistaphylococcal in surgical conditions for 4 to 12 h.

## 1. Introduction

Triclosan (2,4-dichlorophenoxyphenol) is a broad-spectrum antimicrobial that inhibits bacterial fatty acid synthesis. The target receptor in various Gram-positive and negative species such as *Staphylococcus aureus* and *Escherichia coli* is NADH-dependent enoyl-[acyl carrier protein] reductase (FabI) [1,2,3].

Triclosan is an optional adjuvant in absorbable surgical sutures such as polyglactin 910 (Vicryl PLUS “V+”), polydioxanone (PDS PLUS “P+”), and poliglecaprone 25 (Monocryl PLUS “M+”) [4,5,6]. The structural polymers provide different mechanical and absorption properties that drive suture selection given the surgical circumstances [7]. The intended effect from adding triclosan is to inhibit suture microbial colonisation.

In vitro and animal experiments have shown that triclosan sutures (TS) exhibit antimicrobial activity in microorganisms commonly associated with surgical site infections (SSI) such as *S. aureus*, *S. epidermidis* and *E. coli* [8,9,10,11,12].

The expected clinical efficacy is a reduced incidence of surgical site infections (SSIs). Comprehensive meta-analyses of clinical trials have consistently shown a significantly lower incidence of SSIs after wound closure with TS as compared to non-triclosan sutures (NTS) [13,14]. The meta-analysis with the most randomised controlled trials (RCTs) reported a pooled relative risk (RR) of 0.73 [0.65, 0.82], but individual study RRs ranged from 0.09 to 2.45, and only 12% of them were significant. Considering the moderate level of evidence, the WHO made a conditional recommendation to use TS *“for the purpose of reducing the risk of SSI, independent of the type of surgery”* [15].

Pooling can result in a statistical synergy of non-significant study RRs that are in the same direction, whether the cause bias or efficacy. Therefore, the tracing of individual study measures of efficacy to its cause, i.e., exposure to germs *versus* strength and duration of the antimicrobial activity in real-life conditions, is needed.

In vitro studies and experiments in rodents have shown TS antimicrobial activity lasting several days [8,9,10,11,12,16]. However, a literature review has shown that animal models of human surgery often translate poorly to clinical conditions [17]. Moreover, published studies do not relate the antimicrobial activity to triclosan bioavailability on TS.

A review of publication databases found articles on the range of triclosan load on a TS to meet antimicrobial efficacy and non-toxicity objectives, on TS surgical properties, microbiological assays and clinical trial outcomes [8,9,10,11,12,13,14,16,18]. No studies described triclosan bioavailability and pharmacodynamics in the three TS. Product labels indicate the maximum initial triclosan amount (472 µg/m in V+ and 2360 µg/m in P+ and M+), but the average initial amount and variability and the expected duration of bioavailability once implanted are not mentioned. No clinical claims are made [4,5,6].

One literature review reported the initial triclosan load on V+ to be 2.7 µg/cm, but the sources of that value and the methods used to determine it is published [19,20,21].

The use of translational models is unavoidable, given that absorbable sutures cannot be removed from patients for research purposes. Exploring the triclosan bioavailability of triclosan on sutures in animal studies would be scientifically acceptable only if the protocol were accurately specified with in vitro study results, had a significant benefit for patients, and was in compliance with animal ethics, compatible with 3Rs principles [22,23].

Therefore, this study developed an empirical in vitro model specified with ex vivo benchmarks using opportunistically explanted TS fragments from a valid large animal translational model of human surgery in the course of an independent preclinical study according to the *reduction* and *refinement alternatives* of the 3Rs and published approaches to predict drug dissolution in vitro [22,23,24,25].

The critical assumption of this study is that in vivo TS antimicrobial activity reduces the risk of SSI depending on activity duration, with two clinical period phases:-Intraoperative: the objective is to avoid germ inoculation by the suture.-Postoperative: the objective is to prevent suture colonisation while natural defences kill residual microorganisms.

## 2. Materials and Methods

### 2.1. Quantitative Methods

The marginal release rate *R*(*t*) in µg/m/h was defined as the amount of triclosan released per unit of suture length and per unit of time (1):(1)$$R\left(t\right)={\left[triclosan\right]}_{t+dt}\times \frac{V}{L\cdot dt}$$

All rates were derived from the triclosan concentration [*triclosan*]*_t+dt_* measured after immersing a segment of TS of length *L* in meters, in a volume of solvent *V* in litres for a duration *dt* in hours, starting at time *t*.

If the initial amount of triclosan per suture length unit was *A*0, the residual amount at time t since the beginning of the immersion was *A(t)*, calculated as the initial amount less the amount released between immersion start and *t* (2):(2)$$A\left(t\right)=A0-\int_{t=0}^{t}R_{model}\left(t\right)dt$$

Triclosan is a solid at temperatures below 55 °C. It is moderately soluble in water (*Cs* = 0.001 g triclosan/100 g of water, i.e., 10 µg/mL) and highly soluble in ethanol (*Cs* > 100 g triclosan/100 g ethanol) [1,26,27]. Triclosan was assumed to dissolve in liquids according to the Noyes & Whitney relationship (3) [28,29]:(3)$$\frac{dM}{dt}\left(t\right)=\frac{S\cdot D}{d}\cdot \left(Cs-Ct\right)$$

$\frac{dM}{dt}\left(t\right)$ is the dissolution rate of the solute at a given time t, *S* is the dissolution surface (m^2^), *D* is the diffusion coefficient (m·s^−1^), *Cs* is the solubility of the solute in the solvent (Kg·L^−1^ or moles·L^−1^), i.e., the maximum concentration of solute the solvent can hold, above which the solute precipitates. *Ct* (Kg·L^−1^ or moles·L^−1^) is the solute concentration in the bulk of the solvent at time *t* (s), and *d* (m) is the thickness of the solute’s diffusion layer equal to the concentration gradient. Assuming the dissolution surface was constant for each suture during the assay, solubility and diffusion layer thickness were the two independent parameters to modify the dissolution rate.

### 2.2. Standardised Sample Preparation and Triclosan Concentration Determination

The studied sutures were V+, P+, and M+ calibre USP 2-0 (diameter: 0.35 to 0.4 mm), USP 0 (diameter: 0.4 to 0.5 mm) and USP 1 (diameter: 0.5 to 0.6 mm). All test sutures were unpackaged and cut as 35 cm long segments and immersed in 10 mL solvent. All immersions were in glass tubes and incubated at 37 °C in the dark to avoid photodegradation [30]. The tested solvents were pure water obtained by distillation or water plus absolute ethanol as a cosolvent. Immersions were either static or continuously stirred with a tube rotator/incubator (Roto-Therm™ model H2020, Benchmark Scientific, Sayreville, NJ, USA). An independent laboratory accredited for compliance with the EN ISO 17025 standard determined the triclosan concentrations [31,32]. The determinations were performed by liquid chromatography-mass spectrometry using direct sample injection into a high-performance liquid chromatography device (HPLC Shimadzu Nexera X2, Kyoto, Japan) with liquid chromatography-tandem mass spectrometry (LC-MS/MS Sciex 5500, AB Sciex Pte. Ltd., Singapour). The targeted quantification limit was 0.25 µg/L, but the validated limit was 0.01 µg/L.

The samples were prepared at Bordeaux University, Bordeaux, France, and concentrations were determined at Laboratoires des Pyrénées et des Landes (LPL), Mont De Marsan, France, who provided the test tubes and distilled water. Two copies of each in vitro sample were prepared in order to have a pair of determinations. Six copies of each ex vivo sample type and corresponding in vitro controls were prepared.

### 2.3. Opportunistic Explanted TS Fragments and Ex Vivo Triclosan Release Benchmarking

The government authorised the preclinical cardiac surgical trial involving three sheep with scheduled sacrifice on postoperative day-seven (DDPP Paris approval number: APAFIS 12534-2017121118586862 v3) and conducted at the IMMR laboratory, Paris, France. Bordeaux University’s clinical pharmacology department solicited IMMR to piggyback that study. The plan consisted in replacing standard NTS used to close the surgical wounds of the animals by TS and to retrieve TS fragments immediately after animal sacrifice. The piggybacking was agreed because suture replacement did not cause bias to the cardiac surgery study, did not cause additional harm to the animals, and met the *reduction* and *refinement alternatives* of the 3Rs. Moreover, this cardiac surgery involved a validated translational model of human cardiac surgery [22,23,33].

The animals were operated on in conditions similar to human surgery. The surgeons closed the deep muscle layer with V+ 1 and the subcutaneous tissue with V+ 0. The explanted suture fragments were immediately packaged in biohazard bags, one per animal and tissue, labelled, deep-frozen, and forwarded to Bordeaux University, Bordeaux, France.

The explanted intramuscular V+ 1 fragments were long and allowed for the cutting of six 35 cm fragments to be immersed in 10 mL of water. The explants had lost much of their stiffness compared to new sutures.

The explanted subcutaneous V+ 0 fragments were much shorter and were grouped to form pairs of equal length: four groups reached 22 cm and were immersed in 6.3 mL of water, while the two others reached 17.25 cm and were immersed in 4.95 mL, thus ensuring the same length to volume ratio in all samples.

The immersion was static and lasted 12 h. Fragments were then removed, and the tubes with the solvent were forwarded for triclosan concentration determination.

### 2.4. In Vitro Triclosan Release Controls

Segments of V+ 2-0, P+ 2-0, and M+ 2-0 were unpackaged and immersed in static water. Early marginal release rates were calculated with triclosan concentrations determined after segment immersion in a single tube at 4, 8, and 12 h. Subsequent marginal release rates were estimated after segment pre-immersion in tubes during 1, 2, 3, 4, 5, 10, 15, 20, 25, and 30 days. After pre-immersion, each segment was transferred to a tube for the final 12-hour static water immersion used to calculate the marginal triclosan release rate. Segments V+ 0 and V+ 1 were pre-immersed for seven days, then transferred to the final tube for 12 h. Segments were removed from the final tubes before forwarding the tubes for triclosan concentration determination. After 30 days in water, sutures remained stiffer than sutures explanted on day-seven from animal muscle.

### 2.5. Benchmark-Driven Accelerated In Vitro Triclosan Release Models

In vitro conditions were modified targeting a 15-fold to a 60-fold acceleration in triclosan dissolution rate given the two parameters identified with the Noyes & Whitney formula [24]. It was assumed that minimising diffusion layer thickness (*d*) and increasing the difference between solubility and bulk concentration (*C_s_—C_t_)* would have multiplicative effects. The thickness was minimised by continuous rotation at 24 rounds-per-minute (rpm) and 37 °C incubation targeting a six-fold acceleration. The rotation speed was selected based on approaches described previously and was confirmed by checking that stirring dispersed triclosan flakes temporarily when triclosan concentration was equal to 60 µg/mL, i.e., six times triclosan solubility in water at 20 °C [26,27,34,35]. Triclosan solubility was increased by adding 1.2 g of absolute ethanol (EtOH) as a cosolvent to 9 g of pure water (i.e., a 13.3% *w*/*w* solvent). That ratio targeted a 10-fold increase in solubility because preparatory stepwise gravimetric titration showed that that ratio dissolved precipitated flakes of triclosan in a 100 µg/mL saturated solution. A parallel 2.5-fold increase in solubility was targeted with one-quarter of the amount of EtOH (0.3 g) in 9 g of water, i.e., a 3.3% *w*/*w* solvent. The choice of EtOH, among other organic solvents with high triclosan solubility such as acetone, was guided by published experience and by EtOH compatibility with the suture material [36,37]. Another effect of the organic solvent was stabilising triclosan flake dispersion by rotation.

V+ 0 and V+ 1 segments were immersed in parallel sequences of 11 tubes filled with 13.3% *w*/*w* solvent in the rotator/incubator. Immersions were from baseline to 4 h, then 4 to 12, 12 to 24, and then every 12 h until tube 11 from 168 to 192 h. Thus, the total immersion time was eight days. Parallel assays were conducted with similar segments in the 3.3% *w*/*w* solvent.

### 2.6. Estimation of the Initial Triclosan Load

The initial amount of triclosan on a suture segment was estimated with the assumption that it would completely dissolve in a 100% *w*/*w* EtOH/water solvent (1 g EtOH + 1 g water) at 37 °C with continuous stirring for six days. The released concentrations were determined with V+ 0, V+ 1, V+ 2-0, P+ 2-0, and M+ 2-0.

### 2.7. Release Kinetics—Pharmacodynamics Matching Tests

In vitro time-kill analyses were performed at Université de Bordeaux, Aquitaine Microbiologie, UMR 5234 CNRS, 33000 Bordeaux, France, according to CLSI standard M26-A [38]. Suture segments were pre-immersed in the accelerated conditions described previously to establish the relationship between the marginal triclosan release rate and the corresponding antimicrobial activity on triclosan-sensitive bacteria [39]. The tests were conducted at Bordeaux University’s experimental microbiology laboratory. All tests were performed in a safety cabinet in sterile conditions. V+ 0 suture segments of the same batch as previously underwent accelerated triclosan release in 3.3% and 13.3% *w*/*w* solvents, respectively. Pre-immersion durations were 1, 4, 12, 24, and 48 h. At the end of the immersion period, segments were rinsed in distilled sterile water with vortex and then left to dry in a sterile environment. Each segment was then transferred into an Eppendorf tube filled with 10 mL of tryptic casein soy broth (TSB, Difco, BD Diagnostics, Sparks, MD, USA) for a suspension culture with a 0.5 McFarland microbial inoculum (i.e., 10^5^ to 10^6^ colony-forming units CFU/mL) of reference *S. aureus* ATCC 29213 incubated 24 h at 37 °C. Microbial concentration (CFU/mL) in the broth at 24 hours was compared with baseline. Culture tubes were vortexed with the segment inside; next, a 100 µL broth sample was drawn with an aliquot. The sample underwent five serial 1/10 *v*/*v* dilutions. The original broth and each dilution were each spread on an Agar plate with a Mueller Hinton (MH) culture medium and then incubated at 37 °C for 24 h. Viable colonies were counted on the plates at the end of the incubation. The plate selected for determining CFU concentration in the broth was the one where colonies were large enough to be identifiable and individualised. The microbial concentration (CFU/mL) in the culture at each time slot was calculated by multiplying the colony count on the plate by the dilution factor. Two copies of the time-kill assays were performed per solvent and pre-immersion duration.

### 2.8. Data Management and Statistical Analysis

The determined concentrations with sample identifiers and preparation details were recorded in a database. Each pair of copies provided a minimum, a maximum, and a mean. Groups of six copies provided 95% confidence intervals (CI). Statistical analysis computed the time plots of the marginal release rates for each type of suture and solvent. Marginal ex vivo release rates and their controls were computed with CI and tested against their respective controls with two-sided t-tests and a type-I error of 5%. Ex vivo benchmark release rates from subcutaneous and intramuscular explanted fragments were compared.

A mathematical model was fitted to in vitro marginal release rates measured in accelerated conditions and the ex vivo benchmark release rates and residual triclosan over time. The purpose of this fitting was to predict the marginal release rate at a given moment, such as when the antimicrobial activity stopped. Model fitting was performed by single and double exponential decay functions for each series of in vitro triclosan marginal release rates with V+ 0 and V+ 1 after immersion in 13.3% *w*/*w* and 3.3% *w*/*w* solutions, respectively. A similar fitting was performed on 30-day water immersed reference sutures to find out if the model type varied across sutures and triclosan dissolution conditions. The model parameters were solved for each suture/solvent combination. The model with the adjusted coefficient of determination (*R*^2^*_adjusted_*) closest to 1 was selected as the best fit (4) [40]:(4)$$R_{adjusted}^{2}=1-\left(1-\frac{\sum_{i=1}^{n}{\left(y_{i}-{\hat{y}}_{i}\right)}^{2}}{\sum_{i=1}^{n}{\left(y_{i}-\bar{y}\right)}^{2}}\right)\frac{n-1}{n-\left(k+1\right)}$$

The single-decay model estimated the release rate at time t since baseline *R*(*t*) with *R*0 is the baseline release rate and *L* is the half-life (5):(5)$$R_{predicted}\left(t\right)=R0\cdot e^{\frac{-t.\mathrm{Ln}2}{L}}$$

The double-decay model decomposed the single decay baseline rate and half-life into an early decay with a partial baseline rate related to a short half-life overlapping with a later decay with another partial baseline related to a long half-life (6) [41]:(6)$$R_{predicted}\left(t\right)=R_{1}\cdot e^{\frac{-t.\mathrm{Ln}2}{L_{1}}}+R_{2}\cdot e^{\frac{-t.\mathrm{Ln}2}{L_{2}}}$$

The models that best predicted the animal ex vivo benchmarks were selected.

The observed residual amount of triclosan on the suture at time *t* was the extracted initial amount less the sum of the observed released amounts at each time slot since immersion started to time (7):(7)$$A_{observed}\left(t\right)=A0_{observed}-\overset{t}{\underset{t=0}{\sum }}A_{observed}\left(t\right)$$

The predicted residual amount at time *t* was the extracted initial amount less the integral of the fitted model from time 0 to *t* (8):(8)$$A_{predicted}\left(t\right)=A0_{observed}-\int_{t=0}^{t}R_{predicted}\left(t\right)dt$$

Data management and computations were performed in Stata 17, StataCorps LLC, College Station, TX, USA. Mathematical modelling was performed in Maple 2021.1, Maple Inc., Waterloo, ON, Canada.

## 3. Results

### 3.1. Marginal Release Rate in Water Control

The initial rate was high and declined during the first day. It was followed by a slow plateau up to 30 days in all suture types (Figure 1, Figure 2 and Figure 3). The function with the best fit for all three sutures was a double exponential decay. The adjusted R-squared ranged from 0.88 to 0.99.

### 3.2. Marginal Release Rate in Ex Vivo Animal Benchmarks

Ex vivo release rates with V+ on day seven were significantly lower than their controls (Figure 4). The ex vivo intramuscular V+ 1 rate was 0.005 µg/m/h [0.003, 0.006] *versus* 0.30 µg/m/h [0.26, 0.35] in control, thus a significant control/explant ratio of 64.2 was found [52.5, 75.9]. The ex vivo subcutaneous V+ 0 rate was 0.015 µg/m/h [0.009, 0.025] *versus* 0.203 µg/m/h [0.179, 0.220] in control, thus a significant control/explant ratio of 13.6 [8.8, 18.5] was realized. The subcutaneous V+ 0 over intramuscular V+ 1 ratio was also significant with 3.13 [1.92, 4.35].

### 3.3. Marginal Release Rate at Accelerated In Vitro Conditions

The shapes of the marginal release rate curves up to seven days of V+ USP 0 and USP 1, in 13.3% and 3.3% solutions, were similar to the shape in water with V+ USP 2-0 up to 30 days. The functions with the best fit were double-exponential decays as well (Figure 5). The initial rates with the 3.3% solution were about 50% lower than with the 13.3% solution, but the two rate functions converged during day-one, and the seven-day rates came close to the ex vivo benchmarks. (Figures for each other suture and solvent combination are in the Appendix A).

### 3.4. Initial Amount of Triclosan on a Suture

The extraction with the three calibres of braided sutures (V+ 2-0, V+ 0, V+ 1) yielded between 10.1% and 11.1% of the 472 µg/m specified maximum amount. In the case of monofilament sutures with a common 2360 µg/m maximum amount, the extraction was 23% of the maximum with P+ 2-0, and 45% with M+ 2-0 (Table 1).

### 3.5. Residual Amount of Triclosan on the Suture over Time

The residual triclosan amount in V+ 0 was close to 30 µg/m on day-7, i.e., 62% of the extracted initial amount. The residual amount was estimated by subtracting the cumulative release sum from the initial amount. The predicted cumulative release was also used as an alternate method (Figure 6). Overall, the residual amount on day-seven was approximately 93% of the maximum initial load. (Figures for each other suture and solvent combination are in the Appendix A).

### 3.6. Release Kinetics—Pharmacodynamics Matching Results

*S. aureus* grew on the full surface of control plates after spreading samples of cultures without exposure to TS. No colonies grew on plates after spreading samples of cultures exposed to V+ 0 immersed in EtOH/water 13.3% *w*/*w* or 3.3% *w*/*w* during 1 h and 4 h, thus demonstrating the biocidal effect of the triclosan on the suture segment.

Viable colonies of *S. aureus* grew on plates after spreading samples of cultures exposed to suture segments immersed in 13.3% *w*/*w* or 3.3% *w*/*w* solvents between 12 h and 48 hours (Figure 7). These results demonstrated that microbial growth rate increased as the marginal triclosan release rate decreased, as predicted. Moreover, the growth rate was faster in cultures exposed to segments pre-immersed in 13.3% *w*/*w* than in 3.3% *w*/*w*. That difference in rates was consistent with the steeper drop in triclosan marginal release rate with 13.3% *w*/*w* than 3.3% *w*/*w*, at equal immersion duration.

## 4. Discussion

This study achieved most of its goals. The in vitro dissolution model was defined according to approaches to predict drug dissolution in vitro, and they agree with the ex vivo benchmarks from TS [24]. Piggybacking of the planned preclinic study according to the 3Rs was achieved, and the opportunistic suture fragments were the source of the benchmarks as planned [22,23,33]. IMMR confirmed that piggybacking was rarely achieved, either because of confidentiality or incompatibilities between the protocol and the piggybacking objectives.

The common shapes and double-exponential decay functions of the 30-day references in static water with the three sutures V+, P+ and M+ USP 2-0 presented an excellent fit and showed that the release kinetics were very similar. The differences between suture types were in the four parameters of the prediction functions that described the initial high release rate, its rapid decline over one to two days and then a low-release rate plateau for up to 30 days. M+ and V+ had a higher initial rate than V+ and then a more horizontal plateau, which could be related to their higher initial load. The sutures were still reasonably stiff on day-30, and had a uniform surface and a marginal release rate still above 0.5 µg/m/h, which was incompatible with the known hydrolysis of the structural polymers once implanted. M+ loses 50 to 60% tensile strength during the first week, and P+ and V+ lose 20% to 25% within two weeks [4,5,6]. Accelerating triclosan release in vitro in proportions indicated by the ratio of the benchmarks and water controls was necessary to estimate the release kinetics as they occur in operated tissues.

The combination of continuous stirring and increased solubility with ethanol as an organic cosolvent were tested methods, and adapting those methods to the needs of this research was straightforward [24,26,27,34,35,36,37]. The expertise of LPL was instrumental in the success of this study because handling the considerable range of triclosan concentrations in the samples required proven procedures to ensure ad hoc dilution and conservation [31,32].

The similar shapes of the marginal release rate curves and the common double-exponential decay forms between V+ 2-0 in static water and V+ 0 and V+ 1 in EtOH/water solutions with stirring were an important interim internal validation test of the accelerated dissolution models.

One possible explanation for the shape of the triclosan release curves could be the rapid release of surface triclosan and the slow release of most of the triclosan embedded in the polymer and released with the progressive hydrolysis of the biomaterial [42,43].

This study also enables qualitative comparisons with prior studies. In vitro static culture medium models should have triclosan release-kinetics similar to the 30-day water reference in this study. In vivo studies performed with subcutaneous sutures in rodents should have release-kinetics with V+ closer to those or our accelerated models [8,9,10,11,12,16]. The main difficulty in interpreting those studies is translating their results to surgery because of the unavailability of triclosan release data. Planning an external validation of all those studies would have been possible if they had reported triclosan release data along with the observed TS antimicrobial activity.

The prediction functions suggested that the antistaphylococcal activity of V+ was lost when the marginal triclosan release rate was in a transition range between 0.175 µg/m/h and 0.287 µg/m/h. That change occurred when exposing cultures to sutures pre-immersed between 4 and 12 h, with 13.3% or 3.3% solutions. The end of the antistaphylococcal activity coincided with the end of the rapid decline in curves and the transition into the low-level plateau. The total triclosan released was between 1.7% and 5.3% of the extracted amount and between 0.2% and 0.4% of the maximum amount in V+ (472 µg/m) [5]. These results suggest that V+ has an estimated antistaphylococcal activity duration with bactericidal strength for 4 to 12 h postoperatively; covering the intraoperative and early postoperative risks. The residual amount of triclosan after that period is more than 95% of the actually extracted triclosan and more than 99% of the maximum authorised amount on V+.

According to the determinations with the explants from muscle and subcutaneous tissue, most triclosan remaining after that period is released progressively for at least seven days with either bacteriostatic activity or a non-lethal release.

The minimum inhibitory concentration (MIC) of triclosan in *S aureus* is close to the MIC in other common species found in SSIs, including *E. coli* and *S. epidermidis*. Therefore, the antistaphylococcal activity duration measured in the time-kill analyses should predict the antimicrobial activity duration in several other triclosan-sensitive microorganisms often isolated in SSIs with triclosan MIC < 1 µg/mL. Intrinsically, triclosan-resistant species such as *P. aeruginosa*, where the triclosan MIC ranges from 128 µg/mL to more than 1000 µg/mL, can colonise TS intraoperatively [8,9,10,11,12,18,39,44].

The intensive 4 to 12-h antistaphylococcal biocidal-level activity could explain significant differences in efficacy between trials. Differences could depend on the microbial load, microbial nesting at a distance from the suture (e.g., haematoma, devitalised tissue, implant), and the time the patient’s natural defences required in order to kill residual microorganisms.

This study was executed according to the protocol, with some deviations. The main deviation was that ex vivo benchmarks were available only with V+ sutures. Regarding P+ and M+, the study yielded only water controls. In addition, the preparation of a few samples failed, resulting in outliers or missing concentrations. However, those failures were compensated by the backups (at least two copies of each determination) and the small repercussion of an error at a time slot on the next ones.

One potential bias concerning the ex vivo benchmarks was the risk of mechanical removal of triclosan during suture dissection. However, the visual examination of sutures when preparing them for immersion showed that most dissection damages concerned the subcutaneous fragments because of dense adhesion to tissue. Thus, subcutaneous dissection retrieved sutures incompletely and fragmented the available material into segments less than 5 cm long. The same type of suture in muscle had less adhesion, and several fragments were 35 cm long. Determinations showed that ex vivo triclosan marginal release rates in subcutaneous explants were significantly higher than in intramuscular explants. Therefore, dissection in this study had little effect on the triclosan load compared with the differences between muscle and subcutaneous tissue over seven days [17].

The main limitations of this study were the incomplete extraction of the initial triclosan load, no benchmarks before seven days, no explanted fragments with polydioxanone and poliglecaprone 25 sutures, the uncertain of translating sheep benchmarks to humans, and not reproducing suture polymer erosion. The in vitro model focused on triclosan dissolution and did not release the bulk of triclosan that is assumed to be embedded in the polymer. If polymer hydrolysis had been reproduced, the time-kill analysis might have detected antistaphylococcal activity after 12-h of segment pre-immersion.

The difference between subcutaneous and intramuscular implantations show that different models are required to predict TS antimicrobial activity in various clinical situations.

Future studies should explore the amount of residual triclosan.

## 5. Conclusions

This study determined triclosan release rate kinetics and the antistaphylococcal activity duration of polyglactin 910 absorbable sutures. An in vitro dissolution model was developed and specified using ex vivo benchmarks. The dissolution model was continuous, stirring in 3.3% and 13.3% ethanol/water solvents at 37 °C. The benchmarks were the triclosan release rates measured in suture fragments opportunistically retrieved from sheep muscle and subcutaneous tissue after independently planned experimental cardiac surgery.

The exposure of similar sutures to the dissolution model showed an early high release rate that rapidly declined and transitioned to a long-lasting slow plateau after the second day.

A time-kill analysis showed antistaphylococcal activity with bactericidal intensity for 4 to 12 h, and staphylococcal growth with sutures exposed for more than 12 h, with both solvents. The end of the antistaphylococcal activity coincided with the end of the rapid decline in the marginal release rate. Therefore, the residual amount of triclosan was probably more than 99% of the maximum authorised amount on polyglactin 910 braided plus triclosan sutures.

This study showed that polyglactin 910, polydioxanone, and poliglecaprone 25 have similar triclosan release profiles. It also showed a similar profile between polyglactin 910 sutures in static water and an accelerated dissolution model. Differences between suture types and dissolution models seem to be mainly in the maximum initial release rate and the half-life of the decline of that rate.

Sheep are a valid translational model for human cardiac surgery. Polyglactin 910 sutures with triclosan exhibit an intense antistaphylococcal activity for a duration of four to twelve postoperative hours. This activity logically applies to microbial species with a minimal inhibitory concentration of less than 1 µg/mL.

## Figures and Tables

**Figure 1 pharmaceutics-14-00539-f001:**
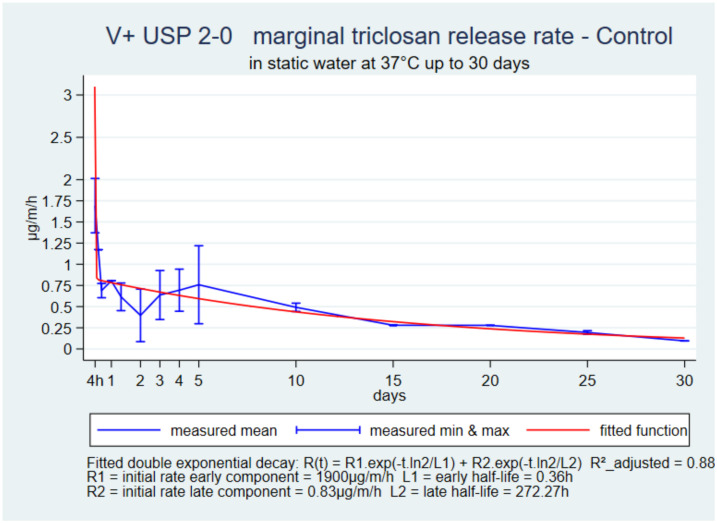
V+ USP 2-0, marginal release rate in static up to 30 days.

**Figure 2 pharmaceutics-14-00539-f002:**
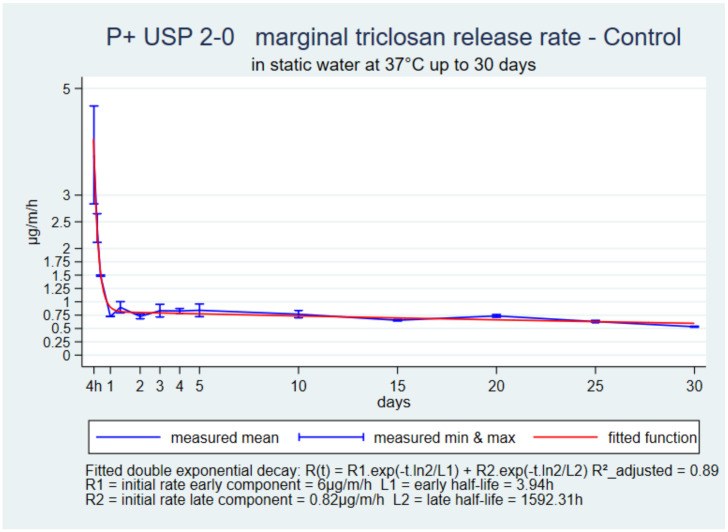
P+ USP 2-0, marginal release rate in static water up to 30 days.

**Figure 3 pharmaceutics-14-00539-f003:**
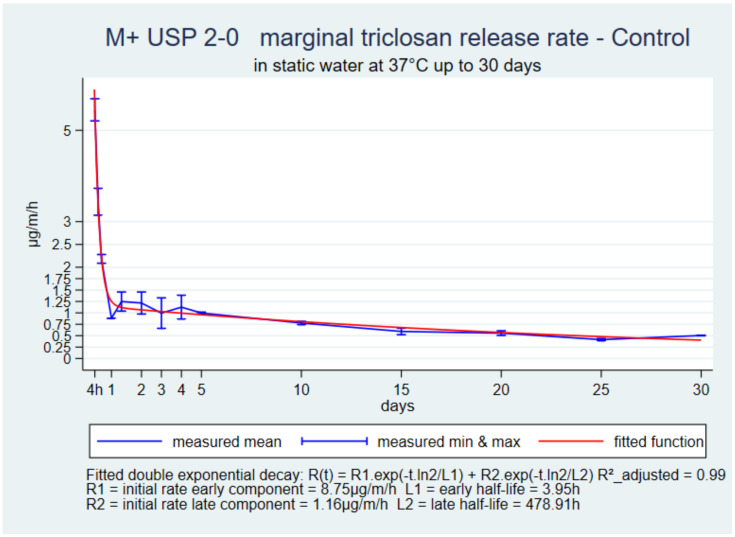
M+ USP 2-0, marginal release rate in static water up to 30 days.

**Figure 4 pharmaceutics-14-00539-f004:**
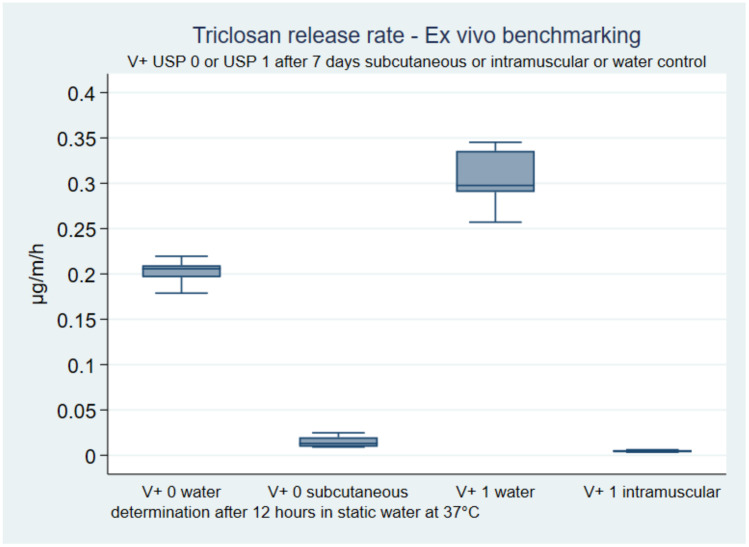
Ex vivo marginal release rate benchmarks *versus* in vitro controls.

**Figure 5 pharmaceutics-14-00539-f005:**
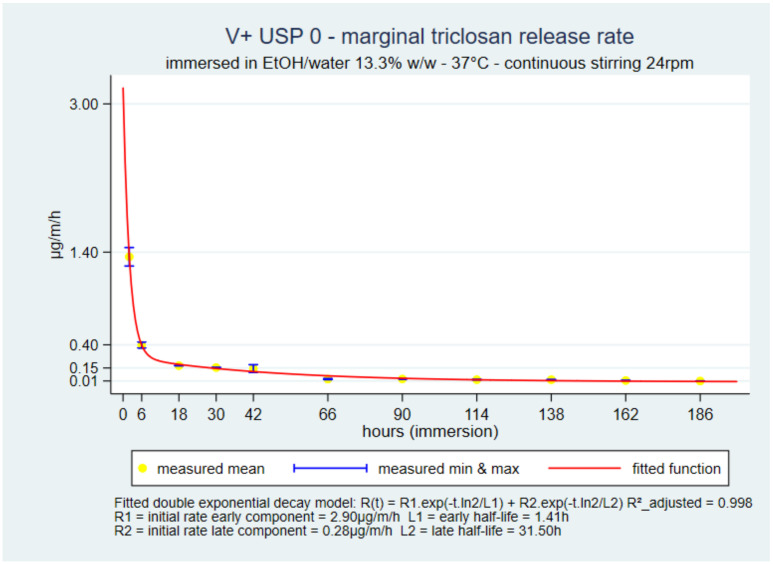
V+ 0 in 13.3% *w*/*w* marginal release rates up to 7 days.

**Figure 6 pharmaceutics-14-00539-f006:**
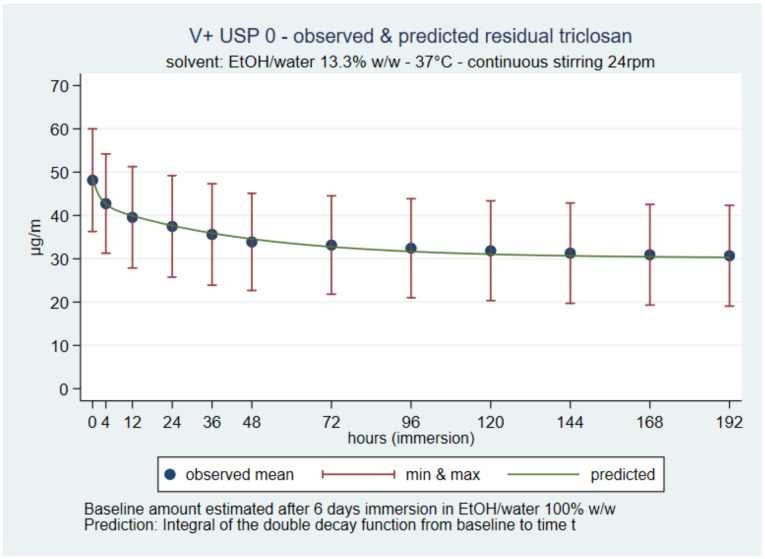
V+ 0 in 13.3% *w*/*w* predicted residual triclosan over time.

**Figure 7 pharmaceutics-14-00539-f007:**
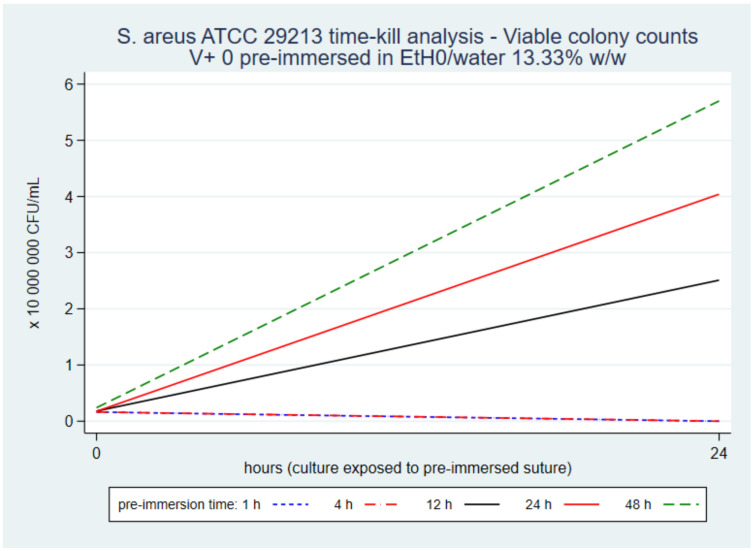
V+ 0 in 13.3% *w*/*w* 24-h time-kill analysis.

**Table 1 pharmaceutics-14-00539-t001:** Estimate of initial amount µg/m.

Release in µg/m	Suture	Days Immersed	N	Min	Mean	Max
Solvent: EtOH/water 100% *w*/*w*	V+ 2-0	6	2	42	47.6	53.1
	V+ 0	6	2	36.3	48.1	60
	V+ 1	6	2	40.3	52.3	64.3
	P+ 2-0	6	2	471.4	551.4	631.4
	M+ 2-0	6	2	971.4	1071.4	1171.4

## Data Availability

The complete datasets used in this study are available in the tables within the Appendix A file: *TriclosanSutures_Pharmaceutics.SupplementaryMaterial.odt*.

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
