# Peer review of "How Long Do Implanted Triclosan Sutures Inhibit Staphylococcus aureus in Surgical Conditions? A Pharmacological Model"

_pharmaceutics, 2022, doi:10.3390/pharmaceutics14030539_

Round 1

Reviewer 1 Report

The authors aim to investigate the release rate and residual triclosan on sutures over time in conditions modeling human surgery to predict the antimicrobial duration. The article has a promising interesting focus on ex vivo-driven pharmacokinetic and pharmacodynamic model of triclosan sutures. The manuscript also possesses a average level of English.

Thus, only major concerns are listed below.

Title: You just work on S. aureus and in title, you write antimicrobial? So if change to antistaphylococcal, I think much well?

ex vivo: it is must be constant italic in all manuscript.

  1. aureus: In abstract, rewrite complete and after that you can used abbreviations. Also, rewrite italic in all manuscript. However, if you start sentence, please write full.

Line 15-18: Rewrite

Line 17-18: The rates combined with estimates of the initial load enabled to calculate residual triclosan over time. This conclusion not methods.

Introduction: Besides brief, the introduction covers most of the bullet points referred to throughout the manuscript, however, antistaphylococcal activity of triclosan are not mentioned; this should be revised. In addition, Introduction need to speak more about the objective of the article.

Line 29: Add Reference

Line 33: Add Reference

Line 35: In vitro Italic (change in all text)

Line 39: .[9, 10] All references written after stop, rewrite in all text before stop point.

Line 44: Add Reference 

Line 51: Add Reference 

Line 54: Add Reference 

Materials and Methods: Almost without references? You need to add some for protocols used.

Line 108: Bordeaux University, Add, France

Line 226: Marginal release rate in ex vivo animal benchmarks, Why table and figure for same results?

Line 288: Moreover, growth rates were faster with segments immersed in 13.3% w/w than 3.3% 288w/w. material). Remove word material and rewrite the sentence.

Line 295-297: Please revise this: Although surgeons select sutures for an operation according to their mechanical properties and absorbability, the availability of optional antimicrobial properties may reduce the risk of SSIs and their consequences.

Discussion: Please Add References (Just contain three References?)

Line 348-349: Why this sentence (This article and its supplemental material provide all the methodological details and raw data to enable reanalysis and carry-on research on this topic.)

References: 24 References is not enough for research article. At least from 35-50.

Author Response

Dear Reviewers,

Thank you for your recommendations to improve the manuscript.

Dr. Goncalves, Pr. Moore, and I have revised the manuscript to follow the recommendations.

We do hope that this upgraded version meets the journal’s standards. The checklist of change requests and answers are in the table below. The line numbers have changed as parts of the text were modified.

Some of the questions could not be answered with a few words only. Those were related to the common release mechanism described by a double-exponential decay in all situations. We supported our answers graphically, replacing Figure 1 with its original three figures. These add key information instead and support/illustrate simple comments.

As for references, we added all the relevant ones. One comment concerned “too old references”. We have added recent references but when the most relevant reference was more than 15 years and without relevant updated, we had no choice but to stick to the old relevant reference.

Thank you

Best regards

Frederic C. Daoud, MD, MSc

The checklist was:

Topic

reviewer 1 (18 Jan. 2022)

reviewer 2 (26 Jan. 2022)

authors' answers

Does the introducdtion provide suffiicent backgroup and include all relevant references?

must be improved

must be improved

updated

is the research design appropriate?

yes

yes

no change

Are the methods adequately described

can be improved

yes

updated

are the results clearly presented

can be improved

can be improved

updated

Are the conclusions supported by the results

yes

yes

no change

General feeling about manuscript

"the article has a promising interesting focus on ex-vivo driven pharmacokinetic and pharmacodynamic model of triclosan sutures"

"the manuscript…is interesting and the authors conduced all the studies thoroughly and discussed the results with the support of literature. Hence, recommended for publication after addressing the following comments"

Comments and suggestions

title

you work on S. aureus and in title your write antimicrobial? So if change to antistaphyloccocal, I think much Well?

Reconsider title of the manuscript

see proposal

general

ex vivo: it must be constantitalikc in all manuscript

done

general

S. aureus In abstract rewrite complete and after that you can used abbreviations. Also rewrite italic in all manuscript. However, if you start sentence, write full

done

line 15-18

re-write

rewritten

line 17-18

the rates combined with estiimates of the initial enabled to calculate residual triclosan over time. This conclusion not methods.

rewritten

Introduction

Besides brief, the introduction covers must of the bullet points covered througout the manuscript, however, antistaphylococcal activity of triclosan are not mentioned. This should be revised. In addition, introduction needs to speak more about the objective of the article.

Need to be elaborated with a background of study and literature

done

Line 29

add reference

done

Line 33

add reference

done

Line 35

in vitro (italic in all text)

done

Line 39

.[9,10] All references written after stop. Rewrite ina all text befroe stop point.

done

Line 44

add reference

details in discussion

Line 51

add reference

done

Line 54

add reference

no reference: this is the innovative core of the research. In order to address the question, more explanation provided

Materials and methods:

Almost without refrences? You need to oadd some of protocols used.

Line 108

Bordeaux University, add France

done

Line 226

Marginal release rates in ex vivo animal benchmarks, Why table and figure for same results?

Resolved: Table 1: moved to supplemental material. Figure 2 matained.

Line 228

Marginal release rates were faster with segments immersed in 13.3%w/w and 3.3%w/w 288w/w. material). Remove material and rewite the sentence.

done

Discussion:

Please add refrerences (just contain three references)

Line 348-349

Why this sentence (This article and its supplemental material provide all the methodological details and raw data to enable reanalysis and carry-on research on this topic.)

removed

References

24 references is not engough for research article. Add at least from 35-50

Most of the authors cited are too old and only two references from the last 5 years. The author should look at recent advancements in literature and cite a few of them

More references added. Old references updated when recent updates available. Old ones kept if no recent update

Fig 1

font size needs to be improved for the convenience of the reader
x-axis units are missing

done

Fig 2

font size below the x-axis should be increased
x-axis units in the figure are missing

done

What could be the explanation for the rapid decline in release rate over the two days followed by a long-lasting slow plateau?

Overall scheme

The author should provide a simple scheme to represent the work of the manuscript to give brief information about the manuscript to read

done

Reviewer 2 Report

The manuscript entitled “How long do triclosan sutures remain antimicrobial once implanted? An ex vivo pharmacokinetics and pharmacodynamic model” by Daoud et al. is interesting and the author conducted all the studies thoroughly and discussed the results with the support of literature. Hence, recommended for publication after addressing the following comments

  1. Introductions need to be elaborated with a background of study and literature
  2. Suggested to reconsider the title of the manuscript
  3. The font size of the text in Fig 1 needs to be improved for the convenience of the reader
  4. Figure 1 x-axis united as missing
  5. The front size of the text in figure 2 below the x-axis should be an increase
  6. X-axis units in the figure are missing
  7. What could be the explanation for the rapid decline in release rate over two days followed by a long-lasting slow plateau?
  8. Most of the authors of the reference cited are too old and only two references from the last 5 years. The author should look at recent advancements in literature and recommend cite a few of them
  9. The author should provide a simple scheme to represent the work of the manuscript to give brief information about the manuscript to read

Author Response

(The authors gave the same response as above.)

Round 2

Reviewer 2 Report

manuscript revised thoroughly and recommended for publications